# "How I wish we could manage such things": A qualitative assessment of barriers to postpartum hemorrhage management and referral in Kenya

**Nora Miller[1], Junita Henry** [1]*, **Kennedy Opondo[1], Lorraine F. Garg[2], Madison Calvert[3], Emma Clarke-Deedler[1,4], Liddy Dulo[5], Emmaculate Achieng[5], Monica Oguttu[5‡], Margaret McConnell[1‡], Jessica L. Cohen[1‡], Thomas Burke[1,2,6‡]**

1 Department of Global Health and Population, Harvard T.H. Chan School of Public Health, Boston, Massachusetts, United States of America, 2 Global Health Innovation Lab, Department of Emergency Medicine, Massachusetts General Hospital, Boston, Massachusetts, United States of America, 3 Sargent College of Health and Rehabilitation Sciences, Boston University, Boston, Massachusetts, United States of America, 4 Department of Epidemiology and Public Health, Swiss Tropical & Public Health Institute, Allschwil, Switzerland and University of Basel, Basel, Switzerland, 5 Kisumu Medical and Education Trust, Kisumu, Kenya, 6 Harvard Medical School, Boston, Massachusetts, United States of America

‡ MO, MM, JLC and TB are joint senior authors on this work.
* junitahenry@fas.harvard.edu

## Abstract

Maternal mortality rates in Kenya have remained high, with the country reporting 342 deaths per 100,000 live births. A major contributor to this is postpartum hemorrhage (PPH), responsible for 40% of maternal deaths in Kenya and the leading cause globally, particularly in low- and middle-income countries. Timely and effective PPH care is crucial; however, challenges arise when referrals between facilities become necessary. Although Primary health care facilities (PHCs) in Kenya oversee many births and are crucial in PPH risk detection and management, they often fall short due to ill-equipped facilities and inefficient referral systems. This study traced PPH patients from tertiary institutions to their initial PHCs. Through qualitative interviews with healthcare providers, we aimed to examine the primary challenges in PPH management and referral decision-making. We found that, in addition to structural gaps, challenges in collaboration and communication between providers from different health facilities, which may also stem from inadequate training, greatly influenced referral efficacy. Our findings are pivotal for maternal health discourse and policy. Importantly, while many solutions focus on structural inputs, our study underscores the importance of communication between facilities in ensuring timely care. Our findings suggest a need for bolstered emergency preparedness, informed clinical decision-making, and strategic interventions where they are most impactful.

**Data Availability Statement:** The intimate nature of our interviews with PHC providers, combined with the small sample size of 14 participants,

heightens the risk of re-identification, despite deidentification efforts. This risk is compounded by the sensitive nature of the health information discussed, obligating us to prioritize participant confidentiality as per our IRB's guidelines, which states that, "if data is shared outside of the certified research team, all personal identifying information will be removed, and responses will be anonymous and only aggregate information will be presented." We only have permission to share aggregate responses. While we cannot make the full data set publicly available, we've included the S1: PHC Provider Interview Guide in our submission to facilitate an understanding of our research approach and the context of our findings.

**Funding:** This study was funded by the Bill & Melinda Gates Foundation (Grant Number: OPP1176156; TB received the award; LFG received salary, ECD, JLC and MM received effort support), The Foundations for Human Behavior Institute (Awarded to JLC and MM), and the Centre for African Studies at Harvard University (Awarded to ECD). The funders had no role in study design, data collection and analysis, decision to publish, or preparation of the manuscript.

**Competing interests:** The authors have declared that no competing interests exist.

## Introduction

Postpartum hemorrhage (PPH) is the leading cause of death and disability among pregnant women worldwide, with the vast majority of mortality occurring in low- and middle-income countries [1, 2]. Despite significant reductions in maternal mortality in Kenya since the establishment of the Millennium Development Goals, progress has stagnated in recent years, and the country continues to have one of the highest rates of maternal mortality in the world, at 342 maternal deaths per 100,000 live births [3]. In Kenya, PPH accounts for approximately 40% of maternal deaths [4].

It is critically important that women experiencing PPH receive timely care before their condition becomes severe and life-threatening. Added challenges occur when a patient cannot receive necessary PPH care at their initial primary healthcare facility and must be referred to a tertiary facility for adequate treatment. A 2017 review of maternal deaths in Kenya found that health worker-related factors were identified in 75% of deaths due to delays in starting treatment, inadequate clinical skills, low levels of monitoring, poor screening for risk factors at the start of delivery, and delays in deciding to refer [4]. Many of these factors are considered avoidable and could be remedied by increased efforts by primary care providers to improve administrative practices, manage clinical emergencies, and rapidly refer cases, as needed.

Healthcare providers at primary health care facilities (PHCs) in Kenya manage a significant portion of births and play an integral role in screening patients for PPH risk factors, actively managing labor to prevent PPH, and making prompt referral decisions when necessary. However, this level of the health system has been found to be the greatest source of delays in PPH management, as facilities are often not equipped to manage complex cases, and referral systems are weak and poorly coordinated [5]. Understanding what the typical experience is like for PHC providers in managing PPH and coordinating referrals can help the Kenyan health system better understand how to support mothers, health workers, and referral coordinators to prevent, manage, and treat PPH and to reduce the risk of maternal mortality.

## Objectives

The Harvard T. H. Chan School of Public Health, the Massachusetts General Hospital (MGH) Global Health Innovation Lab, and the Kisumu Medical and Education Trust (KMET), a non-profit organization in Kisumu, Kenya partnered to assess contextual barriers to PPH management in Kenya. The purpose of this qualitative assessment was to explore the main challenges to management of PPH among providers at the primary health care facility level in Kenya, and to understand how decisions about referrals are made to get women to the care they need. The study was motivated by the following overarching research questions:

1. How did PHC providers manage patients with PPH that were referred to higher-level facilities?

2. What is the typical experience of a PHC provider when managing PPH and the referral process for these cases?

3. What are the main challenges providers at PHCs face in managing and referring PPH cases?

4. What recommendations do PHC providers have for improving PPH management and referral processes?

## Methods

### Study setting

The Kenyan health system is decentralized to the county level, where county governments are responsible for the coordination of health services, including referral [6]. The health system has six levels of care. Level 1 includes community health units and community health workers who provide basic preventive and promotive health services at the community level. These units play a role in disease surveillance and health promotion activities, contributing to the early detection and prevention of outbreaks [5]. Levels 2 and 3 are primary health care facilities (PHCs) that provide basic obstetric care and refer emergency cases to Level 4 or 5 county-level referral health facilities when surgical interventions or blood transfusions are required. Level 2 facilities, often referred to as dispensaries, provide outpatient services, including antenatal care, immunizations, and treatment of common illnesses. Certain Level 2 healthcare facilities perform delivery services; however, they are required to discharge mothers by the end of the day due to the lack of inpatient care services. Level 3 facilities, known as health centers, offer more comprehensive services, including delivery services, basic emergency obstetric care, and inpatient care. These facilities are typically staffed by clinical officers and nurses. Level 4 facility staff include medical officers and occasionally specialists. They manage common medical conditions and minor surgical procedures [6]. Level 5 hospitals, also known as county referral hospitals, offer a wider range of services compared to PHCs, including specialized outpatient services, emergency surgery, and inpatient care for more severe cases. They often have better infrastructure and more specialized staff, such as surgeons and anesthetists, compared to lower-level facilities [7, 8]. The major difference between level 4 and 5 facilities is that Level 5 are teaching hospitals and are better equipped. Level 6 includes national-level referral hospitals which are under the national government and semi-autonomous in their operations, of which there are five facilities across the country [7, 8]. These national referral hospitals provide specialized care, advanced diagnostic services, and are equipped for major surgeries and complex medical conditions. They serve as training centers for medical professionals and are often involved in medical research and the development of clinical guidelines [7].

This study was conducted in three counties–Nairobi, Kisumu, and Kakamega—in western Kenya. There is wide variation between counties in the proportion of births that take place in a facility, ranging from 47% to 89% of deliveries in the three study counties [9]. Each county has at least one Level 5 referral hospital, which serves as the coordinating and referral hospital for lower-level facilities within the county. These referral hospitals have the capacity to provide basic as well as comprehensive emergency obstetric care (CEmOC), including blood transfusion and surgical intervention, while the PHCs (Levels 2 and 3) provide basic emergency obstetric care (BEmOC) [5, 10]. BEmOC includes administration of IV antibiotics, magnesium sulphate, and parental oxytocics as well as performing manual removal of the placenta, removal of retained products of conception, assisting in vaginal delivery (e.g., by vacuum extraction), and performing newborn resuscitation. CEmOC includes all of the BEmOC functions as well as performing caesarean sections, providing emergency obstetric anesthesia, and administering blood transfusions. Providers are trained on and expected to adhere to the Kenyan National Guidelines for Quality Obstetrics and Perinatal Care [5].

### Study design

We employed a qualitative case study and phenomenological approach to understand the experiences of PHC providers managing and referring patients who developed PPH. The case study approach allowed us to review the details of specific patient cases to better understand

exactly which steps were taken at the PHC to identify, manage, and refer for PPH. Building off these cases, the phenomenological approach allowed us to understand the personal and lived experience of providers working at PHCs in Kenya managing PPH referral cases.

## Sampling procedure

From 31st October 2018 to 28th February 2019, we identified patients who had delivered in a PHC and were subsequently referred for PPH management in one of three referral hospitals in Nairobi, Kakamega, and Kisumu counties where the study team was collecting data on post-partum hemorrhage quality of care at referral hospitals [10]. The three referral hospitals were part of a larger study and were purposively selected because they manage high volumes of deliveries (between 17 and 50 per day in 2018) and therefore see large numbers of PPH cases. Patients were recruited into the study at the referral facilities, where they were asked a series of questions about their PHC experience prior to referral. Patients were eligible if they were over 15 years old, had delivered at a PHC and later were referred to the study facility for management of PPH or suspected PPH, and were sufficiently stable to provide consent. Enumerators approached eligible patients once they were stable (and before discharge) to ask for consent to participate. The enumerator requested information on where the patient received care, including the name of the facility, and the date and time at which she arrived at the PHC for delivery, and asked for permission to contact the PHC(s).

## Data collection

Using the information provided by patients, an enumerator visited the facility in person to invite the relevant providers to participate in the study. Interviews were scheduled at times convenient to the provider and followed a semi-structured guide to capture details about the specific PPH case and how it was managed; demographic information such as background and training on identifying and managing PPH, job satisfaction, and motivation; and perspectives on the typical PPH experience at PHC facilities (see the full guide in S1 Text. A snowball sampling technique was used, where interviewed providers identified other members of the care team involved in each case. If another member of the care team was identified, enumerators used that information to follow up with the additional provider and complete the same process outlined above. The overall aim was to interview every provider involved in a patient's care.

## Data analysis

A semi-structured interview guide was developed by study team members at Harvard and MGH. The guide was reviewed and revised by the KMET team in Kenya to ensure the appropriateness of each question for the study context. After the first few interviews, a few additional probes were added to the study guide, including around the coordination of transportation as well as availability and use of data, as these topics emerged from early interviews as potentially important themes.

The interviewers were all trained Kenyan researchers with backgrounds in qualitative methods. Two of the three interviewers also had clinical backgrounds in maternity care. The interviewers participated in a refresher course on research ethics and methods before conducting interviews. Interviews were conducted in English in a private area within the health facility and were audio recorded. Interviews lasted approximately one hour. While we did not predefine saturation, our goal was to follow a case series approach to interview all PHC providers who managed and referred a PPH case to one of the referral facilities during the study period. We therefore stopped data collection once the study period was over.

The recordings were transcribed verbatim by a Harvard research assistant and were checked by a research assistant on the KMET team. All identifying information within the final transcripts was redacted, including facility and provider names.

We used an inductive thematic approach to code and organize the data. The first author read the transcripts multiple times to understand the data and developed an initial codebook without pre-defined assumptions. Codes were then categorized into overarching themes and subthemes before applying the codebook to the entire dataset using Atlas.ti (version 8.4.4). Two of the study authors applied the codebook to 20% of the transcripts independently and discussed discrepancies in coding decisions to ensure consistency before coding the remainder of the transcripts. As new codes emerged, they were discussed by the data analysts, added to the codebook, and applied to all transcripts such that both the codebook and the emerging themes and sub-themes were revised in an iterative process. Once completed, codes were reviewed to identify the most salient themes regarding the challenges that PHC providers experience in managing and referring PPH cases. The dictionary of codes is included in the "S2 Text". We also reviewed deviant findings across all transcripts to assess whether the themes were valid. The themes were shared with subject matter experts on the study team, including medical doctors in the United States and trained Kenyan nurses. Thematic saturation was reached within approximately 60% of analysis, such that the themes identified in the first 8 interviews were consistent with the themes in the remaining interviews.

### Ethics statement

This study was approved by the Institutional Review Boards of Jaramogi Oginga Odinga Teaching & Referral Hospital and the Harvard T.H. Chan School of Public Health (#IRB00047360). Permission to conduct the study was obtained in each study facility. Verbal consents were obtained from each participant after reading a consent form, including patients who consented to their case being reviewed in this study. Interviews with providers were conducted in settings where confidentiality could be maintained. All identifying information was removed from transcripts before analysis, including references to various referral hospitals and other facilities in the geographic area. The audio files were deleted as soon as transcription was complete, per IRB guidelines.

### Inclusivity in global research

Additional information regarding the ethical, cultural, and scientific considerations specific to inclusivity in global research is included in the "S1 Checklist".

## Results

### Sample description

A total of 14 in-depth interviews were conducted with health workers at PHCs. All providers were nurse-midwives. Providers had an average age of 41 years and approximately 16 years of experience, with an average of 8.5 years working in the facility in which they were interviewed. The providers had an average of 4.7 years' experience in their current position and reported attending to approximately 25 deliveries per month. The sample included a range of both early-career health workers and experienced health workers. We provide a breakdown by provider cadre in "S3 Text"). The providers reported that the facilities conducted an average of approximately 75 deliveries per month, ranging from a low of 10 to a high of 200. While we did not collect data on referrals from PHCs, providers estimated that their facilities referred maternity patients between 1–5 times per month for a range of reasons including PPH. All of

**Table 1. Provider and facility characteristics.**

| Provider characteristics (N = 14) | Mean (range) |
|---|---|
| Age | 41 [26,59] |
| Years of experience | 16 [1, 37] |
| Years in facility | 8.5 [.5–28] |
| Years in current position | 4.7 [.5–10] |
| # of deliveries personally attended to per month | 24.5 [2, 40] |
| PHC Facility Characteristics (N = 13) | Mean (range) |
| # of patients seen per facility per month | 354 [70, 800] |
| # of deliveries per facility per month | 74.5 [12, 200] |
| Facility open for day and night shifts | 100% |

**Note:** In 13 patient cases, only one provider was interviewed to understand their care and referral experience. For one of the patient cases, we interviewed a second nurse who provided care for that patient.

the facilities were open for both day and night shifts, though the number of staff on duty was typically lower for night shifts versus day shifts. The PHCs cared for a variety of health conditions outside of maternity care, including preventative and other primary care functions. On average, the facilities saw a total of approximately 350 patients per month, ranging from a low of 70 to a high of 800. Table 1 presents a summary of the provider and facility characteristics in this study.

## Main results

Two key themes emerged from the data to describe the challenges providers face in managing and referring PPH cases at primary care facilities. The first set of challenges were related to the identification and management of PPH at PHCs, which ultimately necessitated referral to higher levels of care. Within this theme, important sub-themes included limitations in facility infrastructure, limitations in provider knowledge, training, and experience, and patient-level factors that made the provider's ability to identify and manage PPH more challenging.

A second key theme included barriers to effective and timely referrals once a decision had been made to transfer the patient. At this stage, providers faced internal delays in decision-making, interpersonal and communication challenges between facilities, structural barriers to efficient referrals, and additional patient-level factors that influenced the timeliness of referrals. These themes and subthemes are described in more detail below.

**Key Theme 1: Challenges in PPH management at primary care facilities.** PHCs are readied to provide basic emergency obstetric care, including actions to prevent or treat PPH such as administration of oxytocic drugs, manual removal of the placenta and other products of conception, and monitoring for and treating blood loss [9]. However, the providers we interviewed cited numerous challenges in identifying and managing PPH, including facility-related limitations (Table 2), gaps in provider knowledge and training (Table 3), and other patient-level factors that made PPH identification and management difficult (Table 4).

*Sub-theme 1.1*: *PPH management—limitations in facility infrastructure*. Providers attested to limited staffing, especially during night and weekend shifts. On average, there were 2–3 nurses on staff during the day, and only one nurse on duty at night in each facility. Many providers found it difficult to adequately address patient needs due to limited staff, especially when emergencies arose (Table 2, Quote 1). Some providers noted that staffing in the maternity ward was ad hoc and a provider was only assigned to the ward if a woman was in labor. In severely understaffed facilities, a single provider covered the entire facility on their own,

**Table 2. PPH management—limitations in facility infrastructure.**

| | Illustrative quote |
|---|---|
| 1. | *There are times you may have a client. . .and we found that the nurse was alone and normally when there is a problem you may need somebody for help [0112].* |
| 2. | *She was on the floor and we could not manage to carry the mother to the bed. The mother was already in shock. I went to the gate to ask for help. I told the [security guards], 'please come, just come don't mind about the nakedness of the mother. We have to carry this mother put her on the bed with the legs raised'. . .so that's what we did [0116].* |
| 3. | *How I wish we had staff in the maternity unit to be managing such things, then we would not be referring. If we have other cases, we would manage them here. . . If we had a MO, we would not have to refer them [0118].* |
| 4. | *Now they are doing something about it (getting water in the facility). At least now they are fixing some taps and we will be having water. Sometimes gloves. . .gloves, gloves, gloves, is a challenge. Even Oxytocin sometimes they go and borrow [0099].* |
| 5. | *After giving medication the bleeding subsided, then the doctor prescribed that the mother had to be transfused. We had a problem getting blood in this facility so one of our staff went to [another facility] to get blood from there. We only got one pint of blood [0105].* |
| 6. | *We borrow from even other facilities, if we don't have any, we can borrow from other facilities [0111].* |

resulting in high workloads and limited ability to prevent emergencies such as PPH. Providers in these situations sometimes called for help from security guards, janitors, or other staff in the facility (Table 2, Quote 2).

In addition to low levels of staffing, facilities lacked higher-cadre staff to advise on and provide support in emergencies. Some providers suggested that they would benefit from having a Medical Officer (MO) on duty at all times in order to comply with Kenyan clinical guidelines that suggest nurses should consult MOs if they are unsure of how to treat specific patients. In some facilities, an MO was always available either in-person or by phone. In others, MOs were difficult to reach and were rarely available for in-person consultations. One nurse commented that they would make fewer referrals if they had more qualified staff on duty, including MOs (Table 2, Quote 3).

Supply shortages were relatively common. Distribution of supplies is managed by the Kenya Medical Supplies Authority (KEMSA), with budgets disbursed to each facility by their county government to purchase the supplies they need. Five of the fourteen interviewed providers cited challenges with limited supplies of gloves, sutures, and IV fluids, as well as some medications such as misoprostol and antibiotics (Table 2, Quote 4). Most of the facilities in this sample were not equipped to provide blood transfusions. However, one facility mentioned that they could transfuse but were out of blood when dealing with this particular emergency and had to borrow from another facility before transferring the patient (Table 2, Quote 5). In

**Table 3. PPH management–limitations in provider knowledge, training, and experience.**

| | Illustrative quote |
|---|---|
| 1. | *You explain to the mother to empty the bladder frequently and then to change the pads. And then you monitor those pads, how often she changes. If there is heavy bleeding, she can report it to the office [0113].* |
| 2. | *I was just monitoring the vital signs for the mother and the baby and the labor progress. . .I didn't really think, even if she usually had PPH previously, that it would become that serious. I thought it would be something that would be manageable [0099].* |
| 3. | *That is just approximation because some blood can be on the floor. You cannot tell the exact amount; it is just approximation [0105].* |
| 4. | *Maybe the [lack of] knowledge would be not on identification but maybe on management. Maybe there could be some delay in giving the first oxytocin, I don't know. Maybe someone would give it after she has done every other thing then she gives the oxytocin later [0099].* |
| 5. | *We need to be trained more about new updates and also any other knowledge that is related to active management [0105].* |

**Table 4. PPH management—patient-level factors leading to delays in PPH identification and management.**

| | Illustrative quote |
|---|---|
| 1. | *Some of them come in delayed or delay coming to the hospital for management and then some of them haven't attended antenatal clinic so it's hard to detect any problems antenatally [0103].* |
| 2. | *I gave her ten units [of oxytocin], I massaged the uterus, I examined. There were no tears. But then she told me she usually has PPH whenever she delivers. So, that scared me already. She has had history of PPH and now she has high BP. So now I was anticipating she may have PPH because of those two things [0099].* |

Kenya, PHCs are not typically expected to have blood transfusion capacity. Providers stated that if a shortage of medication or supplies occurs, either the patient or provider must pay out-of-pocket for the necessary supplies at a local pharmacy. In some cases, as one nurse described, facilities rely on other facilities to fill commodity gaps (Table 2, Quote 6). This practice can lead to additional delays in care.

*Sub-theme 1.2*: *PPH management–limitations in provider knowledge, training, and experience.* In general, PPH was perceived to be rare, and providers had limited hands-on experience managing PPH patients. This lack of experience influenced the level of confidence providers felt in being able to identify causes of PPH and describe the steps that should be taken to manage excessive bleeding. For several less-experienced providers, the patient under discussion was among the first PPH cases that they had managed on their own. These providers described patients' bleeding as being 'uncontrollable' or 'too far along' and outside of their skill set, leading to the decision to refer.

Monitoring of patients in the postpartum period is a critical action for early identification and management of PPH [5]. However, monitoring was relatively infrequent in this sample and often the responsibility for monitoring blood loss was placed on the mother herself, who was told to alert staff if she was bleeding heavily (Table 3, Quote 1). Providers noted that they were aware of guidelines for the recommended frequency of monitoring (e.g., hourly) but their description of the PPH cases suggested that adherence to these guidelines was irregular. Low levels of monitoring led to catching PPH when it was already well-established and challenging to manage. Even when providers were aware of PPH risk factors for specific patients, they described being surprised by the development of excessive bleeding (Table 3, Quote 2).

Quantifying blood loss was highlighted as a key challenge in diagnosing PPH. While most providers defined PPH as 500 ml or more of blood loss after delivery, they mentioned that they often struggled to tell how much blood had been lost, relying on gauze pads, the amount of blood on the sheets or floor, or the alertness of the patient to assess the severity of the situation. A nurse described this estimation method as being an unreliable source of information on the mother's condition (Table 3, Quote 3). A more experienced nurse highlighted the need for additional data points such as skin pallor, patient history, and other considerations that might indicate whether the patient needed additional steps for PPH. These data points, however, were learned through experience managing PPH cases, which not all providers had.

Providers suggested that gaps in knowledge are greatest regarding how to manage PPH once it has started (Table 3, Quote 4). Another nurse suggested that trainings were outdated, and they needed information on the latest practices for PPH management (Table 3, Quote 5). Providers stressed that trainings should be hands-on and practical so that providers gain confidence in their decision-making during emergencies.

*Sub-theme 1.3*: *PPH management—patient-level factors leading to delays in PPH identification and management.* Providers additionally highlighted factors at the patient level such as delays in seeking antenatal or delivery care that made it challenging to screen for, prevent, and manage PPH. When patients arrive at a facility in later stages of labor (i.e., needing to go

straight into delivery) and without any antenatal screening, providers are often unable to check for PPH risk factors or refer them immediately to a more suitable facility (Table 4, Quote 1). In one case, it was only in the postpartum period when a patient told the nurse that she had a history of bleeding in her six prior deliveries (Table 4, Quote 2). Providers felt that better antenatal screening and more information on their patient's histories and risk factors would improve their ability to prevent and manage PPH.

**Key Theme 2: Barriers to effective and timely referrals.** The Kenyan guidelines for obstetric and perinatal care stress the importance of timely referrals in cases where the first facility is unable to provide adequate care [5]. Providers highlighted some of the challenges they face in coordinating these referrals, including internal delays in decision-making, interpersonal and communication challenges with receiving facilities, structural barriers such as lack of transportation and poor road quality, and patient-level factors that influence the referral process. These factors lead to delays in care for patients with PPH, sometimes putting the patient at risk of serious complications or death.

*Sub-theme 2.1*: *Referral challenges—internal delays in decision-making.* Slow decision-making by providers was recognized as a source of delays for referrals. In some cases, providers suggested that a referral should have been made immediately rather than accepting patients with significant risk factors (Table 5, Quote 1). In other cases, there was slow decision-making about referral after patients had delivered and were recovering in the postpartum ward (Table 5, Quote 2). Providers described a 'wait and see' approach in many cases before ultimately referring, hoping that the patient would improve on their own (Table 5, Quote 3). In other scenarios, providers inherited a case during shift turnover and were dismayed that their colleagues had not already started the referral process (Table 5, Quote 4).

*Sub-theme 2.2*: *Referral challenges—interpersonal and communication challenges between facilities*. After deciding to refer patients, providers cited interpersonal challenges in the relationships between PHC providers and higher-level referral providers, where providers in referral facilities were perceived to be rude or dismissive of PHC providers. Providers commented that the referral facilities insinuated poor clinical management of deliveries at PHCs (Table 6, Quote 1) One particularly poor working relationship is demonstrated by the account of a nurse who expressed hesitation around calling the referral facility out of fear of poor treatment by the staff (Table 6, Quote 2). However, poor inter-facility communication challenges were not the case in all settings, and some providers described the relationship with the referring facility as generally positive (Table 6, Quote 3).

**Table 5. Referral challenges—internal delays in decision-making.**

| | Illustrative quote |
|---|---|
| 1. | *Okay, this mother reported in the morning. We were given a report that there was a mother in labor. She was 6 centimeters [dilated] during the time of delivery. She had come at around 5am and she had high blood pressure. . .. Then we were wondering "Why didn't you refer this mother?" Because with high blood pressure, she may complicate here [0103].* |
| 2. | *The decision was made by the nurses was around 8 p.m. when they noted that the mother was still bleeding, and the referral was done beyond 11 p.m. that shows there was long time after the decision of the nurses who were on duty had decided that the mother should be referred up to the time the ambulance came [0105].* |
| 3. | *I think most delays come now to us, it is this person that maybe takes time to decide whether they should be referred or not. There is this tendency of maybe let's give the patient time, we will manage the patient, she might pick up, so I think that decision-making time becomes a problem [0106].* |
| 4. | *Yes. There were some delays because now this patient delivered around 3pm, so by this 3pm this was the time the admitting nurse. . .identified this mother as PPH, so he ought to have arranged how this mother could be referred for PPH management from 3pm up to the time I received the mother around 6. 3,4,5,6, that span of 4 hours they ought to have referred the mother immediately [0106].* |

**Table 6. Referral challenges—interpersonal and communication challenges between facilities.**

| | Illustrative quote |
|---|---|
| 1. | *Let me say most of them have poor attitude towards this hospital. They complain that we refer most cases, that we can't deliver babies, but I don't think that's case, most of them just complain [0103].* |
| 2. | *You know, sometimes the way these guys handle us, you even fear calling. Yeah, they handle us so badly as if we don't know what we're doing. . . At the [referral facility], we don't know whether they blocked our number, we don't know. It never goes through [0099].* |
| 3. | *The relationship is quite good, we cannot complain as per now. Because they've always been helping us in some difficult situations that the facility has tried to manage. . .When we refer, they receive, and some of the patients— quite a number—have been referred and managed well [0105].* |
| 4. | *Sometimes our referral hospital, [Facility A], sometimes they complicate issues, so we have to call the subcounty RH coordinator to help us in referral and sometimes they say that they don't have a surgeon at the moment. Most of the time they say they don't have the kind of blood like that, so they are requesting to send to [Facility B]. When we call [Facility B] they tell us that our referral hospital is [Facility A], so sometimes we delay referral because of that. We have to call the RH coordinator to assist [0103].* |

Poor working relationships caused providers to have reservations about contacting specific facilities, who would instead call a facility that was geographically more distant in order to avoid communicating with their assigned facility. In other cases, the PHCs would have to call on the sub-county coordinator as an advocate and intermediary to broker the referral, adding further delays to the process as the coordinator negotiated a solution on behalf of both facilities. One nurse described a case in which she had decided to call a facility outside of her referral network to get faster care for the patient. A staff member she spoke to at her non-assigned referral facility responded to their emergency call, but did not want to accept the patient without involving the sub-county coordinator for permission. A back-and-forth conversation between the PHC, the referral facility, and the sub-county coordinator resulted in a delay in getting the patient to appropriate care (Table 6, Quote 4). In these cases, the sub-county coordinator was described as a critical resource for the PHCs when trying to reach referral facilities.

*Sub-theme 2.3*: *Referral challenges—structural barriers to effective referrals*. Kenyan guidelines suggest that it is the responsibility of the referring facility to be aware of the capacity of the receiving facility to manage the patient before making any referrals [5]. In several cases in this study, providers were refused access for their patient based on claims that the referral facility lacked supplies such as blood, or had limited capacity or staffing. This information was relayed by calling various referral facilities and often meant that the provider made multiple attempts before finding a facility that would accept their patient (Table 7, Quote 1).

Transportation was a persistent challenge in nearly all facilities, including the cost of ambulance and taxi services, lack of vehicle fuel even when an ambulance was present, unavailability of ambulance drivers, and poor road quality that further delayed patients from receiving adequate care (Table 7, Quote 2). While some facilities had their own ambulance on site, many more relied on external services for transportation, such as faith-based organizations, local NGOs, or the Kenya Red Cross. Facilities rarely had enough staff to accompany patients to referral facilities, and would sometimes send the patient alone in a taxi. One provider reported that these challenges are frequent and can cause significant delays (Table 7, Quote 3). In some cases, delays may add several hours to the referral process or may require extensive networking and negotiation on the part of the provider on duty to find a mode of transport (Table 7, Quote 4).

While guidelines were available for how referrals should be coordinated, providers described deviating from these procedures when a case required immediate attention (Table 7, Quote 5). Overall, providers referenced these ad hoc decisions as being necessary to manage the emergency that they were addressing. For example, the protocol for emergencies in one

**Table 7. Referral challenges—structural barriers to effective referrals.**

| | Illustrative quote |
|---|---|
| 1. | *We actually wanted to refer her to [Facility A], but we didn't refer her there because they said, "If the patient has bled so much, we don't have blood for transfusion." [Facility B] also had the same issue, so the only facility we had an option to was [Facility C] [0110].* |
| 2. | *We only have one driver who works one weekday duty and one weeknight duty, and one weeknight off. When he is off, we have to use drivers from the other facilities. So you have to call them and tell them we have a patient. Sometimes getting them is an issue. We have to go through some means to call them, and you call till you get someplace else [0118].* |
| 3. | *We have a vehicle but the challenge we have is fuel. Also, most of our patients come without any money. And another challenge will be that there is not somebody to escort the patient. . ..So then it would mean either the nurse on duty will leave the station to go and then we would have to call somebody even if it is at night to escort the client [0099].* |
| 4. | *I am telling you it took time. . .because of the means of transport, because that was at 8pm the mother collapsed, and it took two hours before we could reach the mother to the hospital. At long last we had now to tell the relatives to contribute money [to pay for transport]. So we communicated with [nearby nursing home], even the overall in-charge, she was the one who was trying to call the owner of the nursing home, though they did not pick the phone, until I went there. I left my colleague and went there myself to talk to [the nursing home]. [I said] Here is a matter of saving life and the ambulance, within no time, it came [0116].* |
| 5. | *Because we were just the two of us on duty, there were no other staff around, so we felt that it was something we had to act on very fast because by going around looking for people it may be too late for this client. She started getting restless and we thought that the faster we made a decision the better for this client. Because it was about saving life [0112].* |
| 6. | *It's not the norm [to send patients by taxi]. Most of the time we want to refer the patient, first we call to the coordinator for the facility, then they ideally should be sending us an ambulance. Then they come and take the patient to that facility, but in this case the administrator saw it wise because the other one would have taken long, so we decided to call a driver who could come immediately, then we sorted the mother [0106].* |

facility is to call the referral facility, ask them to accept the patient, and then wait for the referral facility ambulance to arrive. However, the severity of a specific PPH patient meant that they had to skip these steps and send the patient by taxi to the referral facility without advance notice (Table 7, Quote 6).

*Sub-theme 2.4*: *Referral challenges—patient-level factors influencing referral delays*. At the patient level, providers mentioned that financial constraints and concerns about the distance of certain referral facilities contribute to delays transferring patients to appropriate care. While maternity services are free in Kenya as of 2013, patients often incur hidden costs such as transport charges to and from referral facilities. These costs are not covered by the national maternity policy [6]. Providers recounted experiences in which patients refused referral because they were unable to afford the bus fare to come home afterward (Table 8, Quote 1). In other cases, providers had to bargain with taxi or ambulance drivers to decrease the out-of-pocket patient charges for quality transportation (Table 8, Quote 2).

**Table 8. Referral challenges—patient-level factors influencing referral delays.**

| | Illustrative quote |
|---|---|
| 1. | **Respondent**: *They were not for it [referral]. They tried to resist me, but that was the best option for them.*<br>**Enumerator**: *Why were they resisting?*<br>**Respondent**: *They said it is far and they don't have the money.*<br>**Enumerator**: *To go or to come back?*<br>**Respondent**: *To come back, because when they were being taken in the ambulance is free [0114].* |
| 2. | *We really begged [the ambulance driver] so they took 3000 (Kenyan Shillings) but they were telling me 5000. But we begged and begged because I knew that their ambulance has oxygen, it has everything, but a taxi has nothing [0116].* |

## Discussion

### Main findings

This study revealed two main sets of challenges in management and referral of PPH cases from primary care facilities. First, providers at primary care facilities faced a variety of challenges in identifying and managing PPH. PPH was a relatively rare event, and some of the less experienced providers had limited experience managing PPH cases. Results from the larger study estimated a 9% (CI:7%-11%) rate of suspected PPH among observed vaginal deliveries within the three referral hospitals [11]. However, given that these are referral hospitals, this rate is likely more than what might be seen in the broader population or at the PHC sites in which these interviews took place. In addition, facilities were understaffed, lacked higher-level cadres, and had shortages of key medications and supplies for delivery. There was a general limitation in knowledge and training around PPH, though providers suggested that the main challenges were in knowing how to manage bleeding once it had begun (which extend into interpersonal communication challenges), rather than gaps in knowing how to prevent PPH or to identify excessive blood loss. Finally, patient-level delays in seeking care, either through low antenatal care attendance or delays in arriving at a facility to give birth added challenges for PHC providers.

As a result of these issues, providers referred PPH patients whose needs exceeded their abilities. The second set of challenges arose in the process of coordinating these referrals in an effective and timely manner. Notably, delays in making the decision to refer meant that cases were sometimes kept at the PHC for many hours before action was taken. Once a decision was made to refer, providers met with interpersonal and communication challenges with referral facilities, which often required arbitration by a sub-county coordinator. This tension between PHCs and referral facilities meant that PHC providers sometimes contacted a less convenient referral facility to avoid poor treatment by staff at their assigned referral facility. When referral facilities were willing to take patients, there often were additional delays due to structural barriers such as lack of transportation, supply shortages at referral sites such as blood for transfusion, and shortage of staff to accompany the patient. Finally, patients themselves sometimes resisted referrals based on the high cost of transportation to return home.

Other studies on referrals in low- and middle-income countries have identified similar challenges. For instance, we found that providers delayed making decisions about referrals and chose instead a 'wait and see' approach. Studies of obstetric emergencies in Uganda and Ethiopia have similarly found that providers rely on their personal judgment and experience over clinical guidelines [12, 13]. Many studies have similarly found that low provider knowledge and training on PPH management—and emergency obstetric care more broadly—is a major contributing factor to delays in PPH management and referrals [13, 14]. A 2017 review of maternal deaths in Kenya found that 75% of those deaths were driven by provider-related factors such as lack of training or delays in transferring the patient [4].

Clear communication between health centers and referral facilities is highlighted in global and national guidelines as being critical to smooth referrals [7, 10]. Other studies in the region have found that poor pre-referral communication may lead to multiple additional referrals before patients are ultimately placed in an appropriate hospital [14]. Our study found that communication was sometimes hampered due to interpersonal relationship challenges between facilities, where PHC providers avoided calling their assigned referral hospitals out of fear of being treated poorly. This was not true in all cases, but did add to the many reasons for delayed referrals.

Facility-related challenges such as supply shortages, inadequate staff, lack of ambulance services or funds to cover transportation costs were common, both in terms of properly managing

PPH as well as supporting timely referrals. These issues have been documented elsewhere as being major contributors to delays [15, 16]. In Kenya specifically, an analysis of the most recent Service Provision Assessment found that 42% of facility births occur in a facility not able to offer the full package of basic emergency care functions [17], suggesting that if an emergency should occur, these births might not be properly managed.

At the patient-level, it has been widely shown that delays in seeking care contribute to complications when women reach the facility [18, 19]. However, we found that there are additional delays at the patient-level after she has reaches the first facility and before being referred onward. For example, patients may object to referral because they cannot afford transport home from the referral site. In other studies, researchers have found that patients may also prefer not to visit a referral site due to the perception that higher-level facilities are overcrowded, have long wait times, or offer poor quality care [13, 18]. In our study, no such objections about higher-level facilities were found.

### Strengths and limitations

The strength of our study is in the rich data on challenges not only in PPH management at the primary care level but also in–for the same cases–transfer to higher levels of care. There are several limitations worth noting. First, the patient cases that were examined in this paper may not be a representative sample of the average PPH referral case, as there were some instances in which a patient was not recruited into the study (e.g., if the woman's condition never stabilized at the referral facility). We also did not predetermine saturation in this study to guide our sampling strategy. Instead, we sought to interview all providers responsible for managing and referring a PPH case during the study period. We did, however, find saturation in themes after analyzing approximately 60% of our interviews where the identified themes repeated in the remaining data. Adding additional interviews, in particular from more varied facility types (e.g., more rural) may have added new perspectives on the identified themes outside of our findings for the more urban sample in these counties. The sample size for this study was relatively small (n = 14 interviews). Therefore, the findings may not be generalizable, nor will they be generalizable to geographies outside of the study sites that may face greater challenges in referrals, for example, for very remote locations. Finally, this study focused on the perspective of PHC providers, but improvements in referrals for emergency obstetric cases will require multilevel, comprehensive interventions. As such, further research should be conducted to understand the experiences of both lower- and higher-level providers, patients, health system coordinators and supervisors, as well as policymakers and funders. Lastly, future research should explore the underlying reasons for the delayed transfers and the challenges around transfers in greater depth.

### Conclusions & recommendations for research, policy & practice

While many studies have focused on challenges related to the clinical quality of care within facilities, this study adds to our understanding of systemic issues in PPH management and referrals that are critical to quality care. Our findings highlight that managing and referring obstetric emergencies is a deeply complex issue, with challenging factors including patient-level delays, gaps in training and preparedness and communication challenges among health care providers. Solutions to address these delays will necessarily be multifaceted and must reach many levels of the health system and society. There are several opportunities for improvement that can be made now, such as training primary care providers on emergency obstetrics, ensuring that there are vehicles and fuel available for transport, and working across levels of the health system to improve working relationships and communication.

In addition, providers highlighted low antenatal screening and delayed arrival at facilities as a challenge to PPH management and referral. Antenatal screening can identify potential risk factors for obstetric emergencies and may create opportunities for providers to support patients in choosing a facility that suits their specific needs. However, it is nearly impossible to predict PPH, and so sorting patients based on risk factors may not be sufficient. Instead, the health system must be able to respond to unexpected emergencies such as PPH. Recent global conversations on maternal health have suggested a redesign of the delivery care model, whereby all deliveries would be shifted from lower-level facilities to higher-level hospitals that can manage both normal and emergency deliveries [20, 21]. More research should be conducted to understand the linkages between antenatal screening at lower-level facilities and how that would influence care at the patient's ultimate delivery facility.

Our study found significant challenges in coordinating transfer of patients to appropriate facilities. In particular, the number of comments on poor working relationships across facilities was striking. Future research should assess how widespread this interpersonal communications challenge really is, and test interventions to improve collaboration across various levels of the health system. For example, improved communication and collaboration could lead to sharing of information and guidance on how to prepare patients for transport, leading to improved stability of patients once they reach the referral facility. Second, there is clear need for national investment in and prioritization of emergency transportation such as ambulances and improvements in road conditions. A 2020 study in Bomet county in Kenya found that implementing an ambulance system was a highly cost-effective means of improving maternal mortality [22]. Future research could assess scale-up of this type of intervention.

There is a clear need for more robust research to unpack and diagnose the root causes of delays in PPH management and referral, and to design and test solutions. Our analysis focused on structural and behavioral barriers, and may be limited in its ability to measure contextual factors such as the relationships between patients, providers, and other key stakeholders. We suggest that future research integrate the perspectives of a wide variety of stakeholders to understand the challenges within each level of the health system and how communities can collaborate to reduce maternal mortality.

## Supporting information

**S1 Checklist. Inclusivity in global research.**
(DOCX)

**S1 Text. PHC provider interview guide.**
(DOCX)

**S2 Text. Dictionary of codes.**
(PDF)

**S3 Text. Provider breakdown by cadre.**
(DOCX)

## Acknowledgments

The authors gratefully extend our appreciation and recognition to the staff and management teams at each respective facility, health care providers who participated in this study and shared their perspectives, and all interviewers who conducted interviews.

## Author Contributions

**Conceptualization:** Nora Miller, Kennedy Opondo, Emma Clarke-Deedler, Margaret McConnell, Jessica L. Cohen.

**Data curation:** Nora Miller, Kennedy Opondo, Margaret McConnell, Jessica L. Cohen.

**Formal analysis:** Nora Miller, Kennedy Opondo, Madison Calvert.

**Funding acquisition:** Margaret McConnell, Jessica L. Cohen, Thomas Burke.

**Investigation:** Nora Miller.

**Methodology:** Nora Miller, Kennedy Opondo.

**Resources:** Nora Miller.

**Validation:** Nora Miller, Kennedy Opondo.

**Visualization:** Nora Miller.

**Writing – original draft:** Nora Miller, Kennedy Opondo, Lorraine F. Garg, Madison Calvert, Emma Clarke-Deedler, Liddy Dulo, Emmaculate Achieng, Monica Oguttu, Margaret McConnell, Jessica L. Cohen, Thomas Burke.

**Writing – review & editing:** Nora Miller, Junita Henry, Kennedy Opondo, Lorraine F. Garg, Madison Calvert, Emma Clarke-Deedler, Liddy Dulo, Emmaculate Achieng, Monica Oguttu, Margaret McConnell, Jessica L. Cohen, Thomas Burke.

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
