## [Decision Letter · Decision Letter 0]

31 Jan 2024

PGPH-D-23-02267

“How I wish we could manage such things”: A qualitative assessment of barriers to postpartum hemorrhage management and referral in Kenya

Dear Dr. Henry,

Thank you for submitting your manuscript to PLOS Global Public Health. After careful consideration, we feel that it has merit but does not fully meet PLOS Global Public Health’s publication criteria as it currently stands. Therefore, we invite you to submit a revised version of the manuscript that addresses the points raised during the review process.

Please note that we have only been able to secure a single reviewer to assess your manuscript. We are issuing a decision on your manuscript at this point to prevent further delays in the evaluation of your manuscript. Please be aware that the editor who handles your revised manuscript might find it necessary to invite additional reviewers to assess this work once the revised manuscript is submitted. However, we will aim to proceed on the basis of this single review if possible. 

Could you please revise the manuscript to carefully address the concerns raised?

We look forward to receiving your revised manuscript.

Kind regards,

Steve Zimmerman, PhD

PLOS Staff Editor

Journal Requirements:

2. Please include a complete copy of PLOS’ questionnaire on inclusivity in global research in your revised manuscript. Our policy for research in this area aims to improve transparency in the reporting of research performed outside of researchers’ own country or community. The policy applies to researchers who have travelled to a different country to conduct research, research with Indigenous populations or their lands, and research on cultural artefacts. The questionnaire can also be requested at the journal’s discretion for any other submissions, even if these conditions are not met.  Please find more information on the policy and a link to download a blank copy of the questionnaire here: https://journals.plos.org/globalpublichealth/s/best-practices-in-research-reporting. Please upload a completed version of your questionnaire as Supporting Information when you resubmit your manuscript.

3. Please amend your detailed Financial Disclosure statement. This is published with the article. It must therefore be completed in full sentences and contain the exact wording you wish to be published.

If you did not receive any funding for this study, please simply state: “The authors received no specific funding for this work.

4. We have noticed that you have uploaded Supporting Information files, but you have not included a list of legends. Please add a full list of legends for your Supporting Information files after the references list.

Additional Editor Comments (if provided):

Reviewers' comments:

Reviewer's Responses to Questions

**Comments to the Author**

1. Does this manuscript meet PLOS Global Public Health’s publication criteria? Is the manuscript technically sound, and do the data support the conclusions? The manuscript must describe methodologically and ethically rigorous research with conclusions that are appropriately drawn based on the data presented.

Reviewer #1: Yes

2. Has the statistical analysis been performed appropriately and rigorously?

Reviewer #1: N/A

3. Have the authors made all data underlying the findings in their manuscript fully available (please refer to the Data Availability Statement at the start of the manuscript PDF file)?

Reviewer #1: Yes

4. Is the manuscript presented in an intelligible fashion and written in standard English?

Reviewer #1: Yes

5. Review Comments to the Author

Reviewer #1: I read with great interest this well done qualitative manuscript that will aid in allowing specific work to be done to improve the care of patients with PPH in Kenya and likely beyond.

A few comments below may help the readers have a better understanding of the setting .

Methods:

there is a nice description of CEmOC and BEmOC but in addition please add who the personnel are at each level and especially where you did your interviews. For example are these all nurses and if so what degree of baseline training have they had in obstetrics ( 1 year/ 2 year etc) are there any higher level midwives at these centers? You do mention there is no medical officer. But understanding that many of these nurses may have had only 1 year of training before working here helps frame the problem.

If possible can you briefly state what protocols they use such as WHO or FIGO - or whatever it is they use so the reader has an idea of what their usual response is.Particularly in one of the comments * there is mention oxytocin being given after bleeding has begun- do they not practice active management of the third stage(AMTSL)?

*"Maybe the [lack of] knowledge would be not on identification but maybe on management. Maybe there

could be some delay in giving the first oxytocin, I don't know. Maybe someone would give it after she

has done every other thing then she gives the oxytocin later [0099]."

Why were all interviews in english and not also swahili ? does this skew the data to only the more educated providers ?Please explain

Results:

there is repeated mention that PPH is rare- can you actually give an occurrence rate at this birthing centers.

there is mention of having the mother alert the provider if there is heavy bleeding yet there is also a mention that an hour check is standard. Yet there is no clear explanation why this is not done.The reader can assume it is lack of personnel but in fact it may be a custom that has evolved as normal practice and not a lack of personnel .That is a question to clarify

Discussion:

Though there is an emphasis on communication issues between referral sites, a better understanding of why that happens would be helpful to explore with the participants . Often the reason may be that at the referral hospitals if they accept a pt who subsequently dies because she was premorbid on arrival , the maternal death is charged to their maternal mortality rate and there may be untoward governmental repercussions.If this does not happen in Kenya that would be good to note.Exploring why the providers feel there is this lack of expedite transfers would be important to explore-though I realize this may have to be for a different paper but if you have that data now please add

Though clearly mentioned at the start of the discussion the the providers have difficulty with management and need for training(lines 363-364), later in the discussion this emphasis disappears ( lines 435-437 ) and also in the abstract (lines 11) and centers on interpersonal relationship challenges however do the authors not feel equal emphasis should be made of the need for provider training.

6. PLOS authors have the option to publish the peer review history of their article (what does this mean?). If published, this will include your full peer review and any attached files.

**Do you want your identity to be public for this peer review?** For information about this choice, including consent withdrawal, please see our Privacy Policy.

Reviewer #1: **Yes: **kay daniels

---

## [Decision Letter · Decision Letter 1]

1 Jul 2024

PGPH-D-23-02267R1

“How I wish we could manage such things”: A qualitative assessment of barriers to postpartum hemorrhage management and referral in Kenya

Dear Dr. Henry,

Thank you for submitting your manuscript to PLOS Global Public Health. After careful consideration, we feel that it has merit but does not fully meet PLOS Global Public Health’s publication criteria as it currently stands. Therefore, we invite you to submit a revised version of the manuscript that addresses the points raised during the review process.

Unfortunately, the previous reviewer who assessed your manuscript was unavailable. However, additional reviewers have assessed the revised version and their comments are available below. There are just a couple of requested clarifications from Reviewer 2. However, with respect to the suggested reference, please note that we always recommend that you review and evaluate requested works to determine whether they are relevant and should be cited. It is not a requirement to cite these works. We appreciate your attention to this request.

We look forward to receiving your revised manuscript.

Kind regards,

Marianne Clemence

Staff Editor

Journal Requirements:

Additional Editor Comments (if provided):

Reviewers' comments:

Reviewer's Responses to Questions

**Comments to the Author**

1. If the authors have adequately addressed your comments raised in a previous round of review and you feel that this manuscript is now acceptable for publication, you may indicate that here to bypass the “Comments to the Author” section, enter your conflict of interest statement in the “Confidential to Editor” section, and submit your "Accept" recommendation.

Reviewer #2: All comments have been addressed

Reviewer #3: All comments have been addressed

2. Does this manuscript meet PLOS Global Public Health’s publication criteria? Is the manuscript technically sound, and do the data support the conclusions? The manuscript must describe methodologically and ethically rigorous research with conclusions that are appropriately drawn based on the data presented.

Reviewer #2: Yes

Reviewer #3: Yes

3. Has the statistical analysis been performed appropriately and rigorously?

Reviewer #2: N/A

Reviewer #3: Yes

4. Have the authors made all data underlying the findings in their manuscript fully available (please refer to the Data Availability Statement at the start of the manuscript PDF file)?

Reviewer #2: No

Reviewer #3: No

5. Is the manuscript presented in an intelligible fashion and written in standard English?

Reviewer #2: Yes

Reviewer #3: Yes

6. Review Comments to the Author

Reviewer #2: - Overall:

> consistent use of postpartum hemorrhage (you use postpartum, post-partum, Postpartum)

> please provide more info on the health care system in Kenya

- Abstract: Aims not clear, please elaborate

- Methods:

> seems very complicated to get the results, is there a more legible way to describe this?

> also, how were the hospitals and providers selected?

- Results:

> lines 165-167: Providers estimated that their facilities referred maternity patients between 1-5 times per month, on average, but that referrals and PPH cases were relatively rare occurrences.

This doesn't make sense. How many PPH cases per provider and hospital? Can you provide a table with more detailed data on PPH cases in the respective institutions?

- Discussion/Conclusion: please add the use of NovoSeven as a potential therapy in cases that have to be referred/transferred to another hospital (cite Caram-Deelder C et al Efficacy and Safety Analyses of Recombinant Factor VIIa in Severe Post-Partum Hemorrhage. J Clin Med. 2024 May 1;13(9):2656). This might be another option particularly in transfer cases that can be also administered by any trained personnel.

Reviewer #3: This is a strong revision of a well organized interesting manuscript.

It is suitable for publication.

7. PLOS authors have the option to publish the peer review history of their article (what does this mean?). If published, this will include your full peer review and any attached files.

**Do you want your identity to be public for this peer review?** For information about this choice, including consent withdrawal, please see our Privacy Policy.

Reviewer #2: No

Reviewer #3: No

---

## [Editor Report · Decision Letter 2]

27 Sep 2024

“How I wish we could manage such things”: A qualitative assessment of barriers to postpartum hemorrhage management and referral in Kenya

PGPH-D-23-02267R2

Dear Ms Henry,

We are pleased to inform you that your manuscript '“How I wish we could manage such things”: A qualitative assessment of barriers to postpartum hemorrhage management and referral in Kenya' has been provisionally accepted for publication in PLOS Global Public Health.

Best regards,

Hannah Tappis, DrPH, MPH

Academic Editor